# Changes in Phenolic Compounds and Antioxidant Activity during Development of ‘Qiangcuili’ and ‘Cuihongli’ Fruit

**DOI:** 10.3390/foods11203198

**Published:** 2022-10-13

**Authors:** Huifen Zhang, Jing Pu, Yan Tang, Miao Wang, Kun Tian, Yongqing Wang, Xian Luo, Qunxian Deng

**Affiliations:** College of Horticulture, Sichuan Agricultural University, Chengdu 611130, China

**Keywords:** plum, phenolic substances, antioxidant capacity, gene expression

## Abstract

Plums are widely consumed, contain high levels of phenolic compounds, and have strong antioxidant activity. In this study, the main Sichuan cultivars ‘Qiangcuili’ and ‘Cuihongli’ were used to study the changes in the appearance, internal quality, phenolic compounds, and antioxidant activities during fruit development and the expression of phenolic-compound-related structural genes. The results showed that, during development of the two plums, the total soluble solid and soluble sugar contents were highest at the mature stage. The phenolic contents (total phenol content (TPC), total flavonoid content (TFC), and total flavanol content (TFAC)) trended gradually downward as the fruits of the two cultivars matured, while the total anthocyanin content gradually increased in ‘Cuihongli’. The main phenolic components were neochlorogenic acid, chlorogenic acid, ferulic acid, benzoic acid, rutin, and proanthocyanidin B1. The changes in the DPPH and FRAP scavenging activities decreased with fruit ripening. The antioxidant capacity was positively correlated with the TPC, TFC, and TFAC. In the two cultivars, the total phenols, phenolic components, and antioxidant capacity were higher in the peel than in the pulp. *CHS*, *PAL3*, and *HCT1* may be the regulatory genes related to the accumulation of phenolic substances in the pericarp and pulp of ‘Qiangcuili’ and ‘Cuihongli’. *HCT1* may be an important regulator involved in the accumulation of chlorogenic acid in plums. The changes in the quality of the phenols, phenolic components, and antioxidant activity were elucidated during the development of the main plum cultivars in Sichuan, particularly the changes that provided a theoretical basis for the development of bioactive substances in local plum cultivars.

## 1. Introduction

Fruits and vegetables are basic components of the human diet that provide nutrition, improve disease resistance, and help maintain health. A healthy diet is closely related to the intake of fruits and vegetables based on their high antioxidant activity and preventive effects on chronic diseases, such as cardiovascular disease, cancer, and diabetes [1]. Many studies have reported that the strong antioxidant activities in plant foods are related to their high phenolic contents [2,3,4].

Plums are commonly eaten fruits. Plums are nutritious and rich in vitamins, anthocyanins, and other phenolic substances that help maintain health [5,6]. Studies have shown that phenolic compounds from plums are beneficial to health, exhibiting antiproliferative activity against breast cancer, immune activity, hypoglycemic activity and alleviating age-related cognitive decline [7,8,9,10].

The phenolic compounds in plums are mainly anthocyanins, flavonols, and phenolic acids [11]. Polyphenols are the most important secondary metabolites of plants. Phenolic compounds are not only allelopathic; they are also signaling molecules that play an important role in plant growth, organ formation, flowering induction, stomatal movement, and photosynthesis [12]. Many phenolic compound synthetic pathways have been identified in plants, the most important of which is the shikimic acid–malonic acid pathway. The shikimic acid pathway is important for most higher plants to convert primary substances into secondary substances. Furthermore, phenylalanine ammonia-lyase (PAL), chalcone synthase (CHS), and 4-coumarate-CoA ligase (4CL) are the phenolic biosynthetic genes [13,14]. Some studies have reported on the antioxidant activity of phenolic compounds in grape, cherry, and strawberry [15,16,17], but few studies have been performed on the accumulation of phenolic compounds and the related gene expression in plums during fruit development and maturation.

Plums are abundant and widely cultivated worldwide. In China, plums can not only be eaten directly but can also be processed into by-products, such as fruit juice, fruit wine, sauce, moon cake, etc. [18]. By 2021, China‘s plum planting area exceeded 1,900,000 hectares (hm^2^). Sichuan is a major agricultural region in China. The cultivation area and yield rank fourth, next to Guangdong, Guangxi, and Fujian, China. The crispy plum industry is unique to Sichuan Province, and the main cultivars are ‘Qingcuili’ and ‘Cuihongli’. The pericarp and pulp change from white-green to yellow-green with the growth and development of the ‘Qiangcuili’ fruit. The peel changes from green to red, and the flesh changes from white-green to yellow-green during the development of the ‘Cuihongli’ fruit. In this study, the changes in appearance and internal quality as well as the total phenol content (TPC), total flavonoid content (TFC), total flavanol content (TFAC), total monomeric anthocyanin content (TMAC), antioxidant activity, and the expression of genes related to the synthesis of phenolic compounds in the two cultivars were determined. Understanding the accumulation of phenolic substances during the development of plums is important, as it provides a theoretical basis for the development of bioactive substances in local characteristic plum cultivars.

## 2. Materials and Methods

### 2.1. Plant Materials

The two cultivars ‘Qiangcuili’ (QCL) and ‘Cuihongli’ (CHL), which were planted in Wenchuan County, Sichuan Province (31°28′ N, 103°33′ W), were used as materials. Fruits were collected from six developmental stages of 42, 56, 70, 84, 98, and 112 days after flowering, and these sampling stages were called S1, S2, S3, S4, S5, and S6, respectively. The peel and pulp were separated, immediately frozen in liquid nitrogen, and stored at −80 °C for later determination.

### 2.2. Determination of Appearance Quality

At least 30 fruits per cultivar were transported to the laboratory at each stage. Single fruit weight was measured with an electronic balance (ATX124, made in the Philippines). Hardness was determined with a WDGY-4 fruit hardness tester. The color indices L*, a*, and b* (L* represents brightness, a* represents the red–green difference, and b* represents yellow–blue difference) of each fruit were measured at the equatorial position of the peel and pulp using a colorimeter (CR-400).

### 2.3. Determination of Total Soluble Solids (%), Soluble Sugars (%), Titratable Acidity (%), and Vitamin C Content

The TSS content in the mixed fruits was determined by a handheld refractometer with the unit of %. The SS contents in the peel and pulp were determined by anthrone colorimetry [19]. The SS was obtained from the equation: SS (%) = C × Vt × N × 100/Vs × W × 10^6^ (C is the sugar content obtained by a standard curve, Vt is the total volume of the extract, N is the dilution rate, Vs is the sample volume at determination, W is the sample weight, and 10^6^ is the weight unit conversion). TA was measured by sodium hydroxide titration, and the TA was obtained from the equation: TA (%) = K × C × (V_NaOH_ − V_3_) × V_1_ × 100/W × V_2_ (C is the sodium hydroxide standard titration solution concentration, V_NaOH_ is the volume of NaOH used for titration, W is the sampling amount of the sample, V_3_ is the blank consumption of the NaOH volume, V_1_ is the constant volume, V_2_ is the titration volume), and the Vc content was measured by the Fe^3+^ reduction method [20].

### 2.4. Determination of Total Phenolic Content

The Folin–Ciocalteu method [21] was used to determine the TPC, with slight changes. A 100 μL aliquot of fruit extract was transferred to a centrifuge tube, and 1.5 mL of distilled water and 0.1 mL of a formalin phenol reagent (diluted 1:1 with anhydrous ethanol before use, now available, dark preservation) were added, in turn, and shaken immediately. After 1 min, 1.5 mL of a 20% saturated Na_2_CO_3_ solution was added, and the solution was mixed again. The reaction was carried out in the dark for 2 h. The extraction solution was used as the control. The absorbance was determined by a UV spectrophotometer at 765 nm, and gallic acid was used as the standard (50–1000 mg/L).

### 2.5. Determination of Flavonoid Content

The method for measuring the TFC was slightly modified from the method of Yong [22]. A 200 μL aliquot of fruit extract was transferred to a centrifuge tube and mixed with 1.3 mL of methanol, 100 μL of NaNO_2_ (0.5 M), and 100 μL of AlCl_3_ (0.3 M). After a 5 min reaction, 500 μL of NaOH (1 M) was added, and the absorbance was measured by a UV spectrophotometer at 510 nm. Rutin was used as the standard (20–100 mg/L).

### 2.6. Determination of Total Flavanol Content

The TFAC was determined using the p-DMACA method following Li et al. [23], with slight modifications. A 100 μL aliquot of the fruit extract was transferred to a centrifuge tube, and 1.5 mL of distilled water and 1 mL of a 1% p-DMACA solution were added in turn. After 10 min, the absorbance was measured at 640 nm, and catechin was used as the standard (6.25–200 mg/L).

### 2.7. Determination of Total Anthocyanin Content

The TMAC was determined based on the hydrochloric acid–methanol extraction method [24] with a slight modification. After grinding 0.5 g of sample, 5 mL of a 1% hydrochloric acid–methanol solution was added to a 15 mL centrifuge tube and shaken to mix. The mixture was placed at 4 °C in the dark for more than 20 h and was ultrasonicated for 30 min. After centrifugation at 8000 rpm and 4 °C for 5 min, the absorbance of the supernatant was measured at 530, 620, and 650 nm. The total anthocyanin concentration was obtained from Equation (1):TMAC(nmol/g) = (OD_λ_/ε_λ_) × (V/W) × 10^6^(1)
where OD_λ_ = (OD_530_ − OD_620_) − 0.1(OD_650_ − OD_620_), ε_λ_ is the anthocyanin molar extinction coefficient, 4.62 × 10^4^, V is the total volume of the extract, and M is the sampling weight.

### 2.8. Determination of Phenolic Components

The method of Zhao et al. [25] was used to determine the phenolic components in fruits by high-performance liquid chromatography (HPLC), with slight modifications. The peel and pulp were removed from storage at −80 °C and ground into a powder in liquid nitrogen. Then, 0.5 g of the mixed sample was weighed and placed in a 5 mL centrifuge tube. A 1.5 mL aliquot of the extraction solution (V (methanol)/V (water) = 84:16) was added in the dark and fully mixed. The centrifuge tube was placed in an ultrasonic device for 30 min of extraction, shaken at 250 r/min for 2 h at 30 °C, and centrifuged at 4 °C and 8000 r/min for 15 min. The supernatant was filtered through a 0.22 µm microporous membrane to analyze gallic acid, vanillic acid, caffeic acid, protocatechuic acid, chlorogenic acid, ferulic acid, neochlorogenic acid, benzoic acid, quercetin, rutin, and proanthocyanidin B1. The extract was also prepared to analyze the TPC, TFC, TFAC, and antioxidant activity. An Agilent 1260 liquid chromatograph (Agilent Technologies Inc., Palo Alto, CA, USA) and a Comatex C18 column (5 µm, 46 mm × 250 mm) were used at a column temperature of 30 °C and an injection volume of 20 μL. Mobile phase A was ultrapure water containing 0.1% formic acid, and mobile phase B was chromatographically pure acetonitrile containing 0.1% formic acid. The system was run on gradient elution at a flow rate of 1 mL/min. The elution program is shown in Appendix A. The detection wavelength was 280 nm.

### 2.9. Antioxidant Activity Analysis

#### 2.9.1. Determination of DPPH Scavenging Activity

The DPPH scavenging capacity was determined using the method of Brandwilliams et al. [26] with slight modifications. A 2 mL aliquot of a DPPH methanol solution (6.25 × 10^−5^ M) was transferred to a centrifuge tube, and 100 μL of the sample extract was added. After 20 min in the dark, the absorbance was measured at 517 nm. Trolox was used as the standard, and the final result was expressed as μmol/L Trolox equal antioxidant capacity.

#### 2.9.2. Determination of Ferric Reducing Antioxidant Power (FRAP)

The FRAP was determined using the slightly modified method of Benzie et al. [27]. A 100 μL aliquot of the fruit extract was mixed with 1 mL of distilled water and 1.8 mL of a TPTZ working solution and reacted in a 37 °C water bath for 10 min. The absorbance was determined at 593 nm. The final result was expressed as μmol/L Trolox equal antioxidant capacity.

### 2.10. Analysis of Relative Expression of Structural Genes Related to the Synthesis of Phenolic Compounds

RNA was extracted following the CTAB method of Shu et al. [28]. cDNA sequences of phenolic-synthesis-related genes in plums were downloaded from the NCBI. The related sequences were used to search the *Prunus domestica* genome using local Blast [29]. Subsequently, gene sequences were extracted with TBtools [30], and the primers (Appendix A) were designed using Beacon Designer 7 (Version 7.9, Premier Biosoft, San Francisco, CA, USA), and 18S was used as the reference gene. The SYBR^®^ Premix ExTaqTM II kit (Beijing Qingkexinye Biotechnology, Beijing, China) was used for the real-time polymerase chain reaction analysis. The reaction program included pre-denaturation at 95 °C for 30 s, followed by 40 cycles of denaturation at 95 °C for 5 s and annealing at 58 °C for 30 s. The relative expression levels of each gene were calculated using the 2^−ΔΔCt^ method [31], and each sample was replicated three times.

### 2.11. Statistical Analysis

All values represent triplicates, and the results are expressed as means ± standard error. The statistical analysis was performed using IBM SPSS Version 26.0, SPSS statistical software (SPSS Inc., Chicago, IL, USA), and a one-way analysis of variance was performed to detect differences. A *p*-value < 0.05 was considered significant.

## 3. Results

### 3.1. Changes in the Appearance Index during Fruit Development of the Two Plum Cultivars

The single fruit weights (Figure 1C) of QCL and CHL increased gradually during growth and development. The single fruit weight of QCL was 34.13 g when ripe, whereas that of CHL was 19.34 g. The hardness of the two plums (Figure 1D) decreased gradually with growth and development. The hardness values of the QCL and CHL fruits were 11.84 N and 14.40 N at maturity, and CHL decreased faster than QCL during the early stages.

The colors of the peels of the two plum cultivars changed differently. The peel of QCL (Figure 1A) changed from white-green to yellow-green during fruit development, and the L* value (Figure 1E) trended upward. The L* value of the QCL peel was highest during the S6 period, while there was no significant difference in the L* value of the pulp between the immature and mature periods (*p* > 0.05). The a* value (Figure 1F) was negative throughout fruit development, and the b* (Figure 1G) value was initially stable and then increased.

The CHL peel (Figure 1B) began to change color at S4 and was fully red at S6; the L* value of the pericarp increased first then decreased, and the L* value of the CHL pericarp was highest during S3. The flesh changed from white-green to yellow-green as the fruit grew. The a* value of the peel increased gradually beginning in the S4 period and reached the maximum during the S6 period, which was 22.08. The b* value of the pericarp first increased and then decreased.

### 3.2. Changes in TSS, SS, TA, and Vc Contents during Fruit Development of the Two Plum Cultivars

The TSS (Figure 2A) and SS (Figure 2B) contents in the pericarp and pulp gradually increased, and the TA (Figure 2C) gradually decreased with the growth and development of the QCL and CHL fruits. Both cultivars produced higher SS contents in the pulp and higher TA contents in the peel. Some differences were found between the two cultivars. The SS contents in the peel and pulp of QCL were 12.10% and 13.84% at S6, and the TA values were 0.66% and 0.31%, respectively. The SS contents in the peel and pulp of CHL were 12.74% and 14.13%, and the TA values were 0.77% and 0.32%, respectively. During fruit development, the TSS of CHL was always higher than that of QCL, and the SSC contents in the peel and pulp were similar to those of the TSS and higher than that of QCL. The Vc contents in the peel and pulp of the two cultivars (Figure 2D) trended downward first, then increased, and then decreased again. At maturation (S6), the Vc content in the peel was higher than that in pulp between the two cultivars, and the highest Vc content was in the CHL peel (4.72 mg·100 g^−1^ FW).

### 3.3. Changes in Phenolic Content during Fruit Development of the Two Plum Cultivars

The TPC, TFC, TFAC, and anthocyanins were determined in the peel and pulp of the two plum cultivars (Figure 3). The TPC of the peel and pulp of the two plum cultivars decreased gradually with the growth and development of the fruit (Figure 3A). The TPC in the peel was higher than that in the pulp of both cultivars during fruit development. The TPC in the peel of CHL was always high, while the TPC in the pulp was low. The TPC during the fruit ripening stage was in the order of CHL-P (6.80 mg·g^−1^ FW) > QCL-P (4.24 mg·g^−1^ FW) > QCL-F (2.19 mg·g^−1^ FW) > CHL-F (0.88 mg·g^−1^ FW). The differences in TPC corresponded to the differences in phenolic acids.

The TFC (Figure 3B) decreased during fruit development in the two plum cultivars, and it was higher in the peel than in the pulp. Similar to the TPC results, the TFC of CHL-P (20.82 mg·g^−1^ FW) was the highest, followed by QCL-P (6.62 mg·g^−1^ FW), QCL-F (2.20 mg·g^−1^ FW), and CHL-F (2.18 mg·g^−1^ FW) during the fruit ripening stage. The difference in the TPC was consistent with the difference in the flavonol components.

The TFAC (Figure 3C) showed different trends during the fruit development of the two plum cultivars. The TFAC (Figure 3C) fluctuated greatly, and no TFAC was detected in the flesh of the two cultivars at S5 or S6. The total flavanol contents decreased gradually with the growth and development of the fruit, and flavanols were not detected in the pulp of either cultivar at S5 or S6. Except for QCL-F, the TFAC had the highest S2 content during the early stage of fruit development. The TFAC content in the peel of CHL was higher than that in QCL.

The TMAC (Figure 3D) is widely used as an indicator of the fruit and vegetable developmental stage. The TMAC in the peel and pulp of QCL was low and was similar to the CHL pulp. The TMAC increased significantly in the CHL peel from S4 to S6, consistent with the fruit coloration, and reached the maximum at the full-red stage (S6), which was 585.42 μg·g^−1^ FW.

Therefore, the compositions and contents of the phenolic compounds in the peel and pulp of the two plum cultivars were useful as indicators of pigment accumulation in the peels of the two cultivars. They also reflected the maturity of the fruits.

### 3.4. Changes in the Phenolic Components during Fruit Development of the Two Plum Cultivars

Phenolic compounds have important antioxidant activities. Nine phenolic compounds, which have also been reported in other plums, were separated and quantified by HPLC. Similar phenolic compounds were found in the two cultivars, including six phenolic acids (gallic acid, neochlorogenic acid, chlorogenic acid, vanillic acid, ferulic acid, and benzoic acid), two flavonols (rutin and quercetin) and one flavanol (proanthocyanidin B1) (Table 1).

The contents of the phenolic acid components gradually decreased in the QCL peel and flesh during fruit maturation (Table 1). At the fruit ripening stage, the main phenolic acids in QCL-P were chlorogenic acid and neochlorogenic acid, and the order was chlorogenic acid (7.82 ± 0.37 mg·100 g^−1^ FW) > neochlorogenic acid (5.18 ± 0.32 mg·100 g^−1^ FW) > benzoic acid (3.28 ± 0.03 mg·100 g^−1^ FW), whereas gallic acid was not detected in the flesh. Chlorogenic acid and neochlorogenic acid were the only two components observed in the flesh at the mature stage. In contrast with the QCL results, the main phenolic acids in the CHL peel at the fruit ripening stage were vanillic acid and benzoic acid. The phenolic acids in the peel were in the order of vanillic acid (18.67 ± 1.36 mg·100 g^−1^ FW) > benzoic acid (18.37 ± 0.53 mg·100 g^−1^ FW) > chlorogenic acid (8.07 ± 0.28 mg·100 g^−1^ FW) > neochlorogenic acid (7.95 ± 0.05 mg·100 g^−1^ FW) > ferulic acid (1.34 ± 0.04 mg·100 g^−1^ FW), while neochlorogenic acid (2.10 ± 0.20 mg·100 g^−1^ FW) was only detected in the pulp. Neochlorogenic acid and chlorogenic acid declined first and then increased during fruit development. Benzoic acid decreased first and then increased, and its content was higher than the other phenolic components. The contents of neochlorogenic acid, chlorogenic acid, and benzoic acid decreased gradually in the flesh during fruit ripening. All of the phenolic acid components in the CHL peel, except gallic acid, were higher than those in QCL.

Rutin and quercetin were the main flavonols detected. The contents of rutin and quercetin decreased gradually with fruit development in the QCL peel and pulp, and no quercetin was detected in the pulp. The rutin content in the CHL peel peaked at maturity (17.07 mg·100 g^−1^ FW). Quercetin was not detected in the CHL pulp. The rutin and quercetin contents were higher in the peel of CHL than in QCL.

The procyanidin B1 concentration in the two plum cultivars decreased gradually to maturity. The content of procyanidin B1 during fruit ripening was highest in the QCL peel (65.30 ± 4.59 mg·100 g^−1^ FW), which was much higher than that of CHL (8.02 ± 0.03 mg·100 g^−1^ FW). The content of procyanidin B1 in the flesh of QCH was 6.70 ± 0.35 mg·100 g^−1^ FW, and it was not detected in the flesh of CHL.

### 3.5. Changes in the Antioxidant Capacity during ‘Qiangcuili’ and ‘Cuihongli’ Fruit Development

DPPH (Figure 4A) and FRAP (Figure 4B) were used to evaluate the antioxidant activities of the two plum cultivars during development. The trends in the antioxidant capacity of the peel and pulp of the two plum cultivars were consistent with that of total phenolic content. DPPH peaked at the S1 stage and then gradually decreased, and a similar pattern was observed for FRAP. The DPPH and FRAP results show that the antioxidant capacity of the peel was significantly higher than that of the pulp during the fruit ripening stage in the order of CHL-P > QCL-P > QCL-F > CHL-F.

To explore the relationships between the phenolic compounds and the antioxidant activity of the two kinds of plums, the correlations (Table 2) between the TPC, TFC, TFAC, and TMAC and DPPH and FRAP were analyzed. The results showed that the TPC, TFC, and TFAC were closely correlated with DPPH and FRAP. The plum peel/pulp extracts had strong DPPH and FRAP scavenging abilities for free radicals, which may have been attributed to the high TPC and total flavanol contents.

### 3.6. Analysis of the Development Process and Expression of Related Genes in the ‘Qiangcuili’ and ‘Cuihongli’ Peel and Flesh

To further understand the relationship between the expression of related genes and the development stages of the two plum cultivars (Figure 5), we analyzed the expression of nine genes during the development of the two plum cultivars. The results showed that the expression levels of the genes were low in the pulp. Among them, *PAL1*, *PAL3*, *C4H*, *4CL1*, *4CL2*, *HCT2*, and *HCT3* all increased with the maturation of the fruit, whereas *CHS* and *HCT1* trended downward in the QCL pericarp. The expression of *PAL3* decreased in the QCL pulp. The nine genes showed downregulating and then upregulating trends in the CHL peel. The expression levels of all of these genes, except *HCT1*, were higher during the mature stage than during the young fruit stage. In contrast to the pericarp, the expression levels of the nine genes trended down in the pulp.

### 3.7. Correlation Analysis of Phenolic Content and Related Gene Expression during Fruit Development of ‘Qiangcuili’ and ‘Cuihongli’

To further understand the relationship between the phenolics and the related gene expression, we analyzed the TPC, TFC, TFAC, and TMAC parameters in the peel and pulp of the two plum varieties (Table 3). The TPC, TFC, and TFA contents in QCL-P were significantly positively correlated with the expression of *CHS* and *HCT1* during QCL fruit development. The TPC and TFC in QCL-F were significantly positively correlated with the expression of *PAL3*, *CHS*, and *HCT1*. *PAL3* expression was higher with increased TFAC and TMAC, and *CHS* was higher with increased TMAC. The results showed that the *CHS* and *HCT1* genes were closely related to the synthesis of phenolic compounds during fruit ripening, while the *PAL3* gene was closely related to the synthesis of phenolic compounds in QCL pulp.

During CHL fruit development, the anthocyanin content in CHL-P was significantly positively correlated with the genes important in the phenolic synthesis pathway. The correlation coefficients of *PAL1*, *PAL3*, *C4H*, *4CL2*, and *CHS* were all > 0.744, and the correlation coefficient with *CHS* was 0.988. The TPC in CHL-F was mainly positively correlated with *4CL2*, *HCT1*, and *HCT3*. The TFC was positively correlated with the expression of *4CL1*, *4CL2*, *HCT1*, and *HCT3*. The TFAC was positively correlated with *PAL3*, *4CL1*, *4CL2*, *CHS*, *HCT1*, and *HCT3* expression. The expression levels of the nine genes were extremely low in the TMAC; *HCT* gene expression was the highest in the TAFC, followed by the TPC and the TFC. These results show that the *HCT1* gene was closely related to the synthesis of phenolic compounds during fruit ripening.

As shown in Table 4, the neochlorogenic acid and chlorogenic acid contents were significant in the two plum cultivars. Neochlorogenic acid and chlorogenic acid were significantly positively correlated with *HCT1* in QCL peel and pulp. Neochlorogenic acid was significantly positively correlated with *HCT2* in the CHL peel, and neochlorogenic acid and chlorogenic acid in the pulp were significantly positively correlated with *HCT1*. These results show that the *HCT1* gene is closely related to the synthesis of neochlorogenic acid and chlorogenic acid during the ripening of the two plum cultivars.

## 4. Discussion

QCL and CHL are the two main crisp plum cultivars in Sichuan Province. Different from most plum cultivars in which the flesh softens during fruit ripening, flesh softening is not obvious in crisp plums at harvest, as the flesh maintains a certain hardness and brittleness. The hardness values of the QCL and CHL fruits were 11.84 N and 14.40 N, which were higher than ‘Yanzhi’, ‘Taoli’, ‘Oishiwase’, ‘Furong’, and ‘Jinmi’ [32]. The TSS content and TA of the QCL and CHL fruits trended up, and the TA and Vc contents gradually decreased with fruit ripening. Similar to Huang et al. [33], who studied blackberry and raspberry, the sugar content was highest at fruit ripening, and the reduction in Vc content was similar. The QCL fruit changed from white−green to yellow−green during growth and development, and the L* value of the peel trended upward. Li et al. [34] reported that anthocyanin is the cause of the red color in red fruit. The CHL fruit began to color at the S4 stage and turned dark red at S6, consistent with the accumulation of anthocyanin content in the peel.

Plums are rich in phenols, flavonoids, flavanols, and anthocyanins. The TPC of the plums was 174–375 mg/100 g, second only to blackberry and blueberry. In this study, the TPC of the peel was higher than that of the pulp, and the total anthocyanin content in the CHL peel was the highest, which also corresponded to the color of the fruit. Rupasinghe et al. [35] analyzed the peel color of 20 European plum cultivars and found that the anthocyanin content in dark-purple plums was higher than that in green plum cultivars. Therefore, anthocyanin is the main reason for the red color of plums. Red plums are recommended because of the antioxidant effect of edible anthocyanins.

Previous studies have detected neochlorogenic acid, gallic acid, protocatechuic acid, chlorogenic acid, and vanillic acid as the phenolic components of plums [36,37]. This study detected gallic acid, neochlorogenic acid, chlorogenic acid, vanillic acid, ferulic acid, benzoic acid, rutin, and proanthocyanidin B1 in the peel and neochlorogenic acid, chlorogenic acid, vanillic acid, ferulic acid, benzoic acid, rutin, and proanthocyanidin B1 in the pulp. The main phenolic components were neochlorogenic acid, chlorogenic acid, ferulic acid, benzoic acid, rutin, and proanthocyanidin B1. Cosmulescu et al. [38] showed that the TPC in the peel was 4.5 times that in the pulp and 3.2 times that in the whole fruit. This study also showed more types and higher contents of phenolic components in the peel than in the pulp. Thus, the peel should be further utilized.

The TPC and phenolic components trended down during the development of the two plums. This result was also reported in long-fruited *Ribes* tea [39] and apples [40]. The mechanism leading to the decrease in the phenolic content in fruit may be a reduction in phenolic synthesis, or it may be because phenols are continuously transformed into other substances during development. At the same time, as the fruit weight increases, the phenolic content per unit mass decreases [41]. Phenols are important antioxidants and stress-resistant substances with high contents in young fruits and decreased contents in mature fruits. The bitter taste of phenolic substances is an important source of astringency in young fruit and is a protective mechanism against animal consumption. The phenolic substances in the peel and pulp decrease with the growth and maturity of the fruit; therefore, the astringency of the fruit gradually decreases, the sugar content increases, and the edibleness of the fruit increases, which is conducive to the spread of mature seeds after animal consumption. The decrease in phenolic content is also speculated to be an adaptation during the evolution of the fruit. The anthocyanin content in CHL−P increased as the fruit turned red during the middle and later periods of fruit development, which was similar to the results by Gonzdlez et al. [42], which showed that the anthocyanin content peaks in the pericarp and flesh of four plum cultivars during four different periods.

The antioxidant activity of plums is very prominent among many fruits. Remberg et al. [43] compared European plum, pomegranate, grape, pineapple, and other fruits, and the antioxidant activity of European plums was relatively high, indicating that plums have strong antioxidant activity. Phenols are an important class of antioxidants in plants. The DPPH and FRAP scavenging activities of the QCL and CHL fruits were significantly positively correlated with the TPC, TFC, and TFAC, which was consistent with the results for Blueberry [44]. Total phenols had the greatest effect on the antioxidant capacity, which is consistent with the results of Chen et al. [45]. The antioxidant activity of CHL−P was negatively correlated with the TMAC, indicating that, although the TMAC was an important antioxidant, the antioxidant activity of CHL−P was mainly affected by the TPC.

*PAL* is the first enzyme in the biosynthetic pathway to produce phenols by regulating the conversion of phenylalanine to cinnamic acid [46]. The results of this study showed that the *PAL3* gene was significantly positively correlated with the phenolic contents in pulp. *C4H* catalyzes the synthesis of 4−coumarate from cinnamic acid, which is a regulatory factor in the intermediate step of chlorogenic acid synthesis [47]. In the two plum cultivars, the *C4H* gene was positively correlated with total flavonols and the total flavanols in the peels and was significantly positively correlated with the total anthocyanin content in the pulp, which was consistent with the results of Pu et al. [48]. The *4CL* gene functions in different biosynthetic pathways, resulting in different phenylpropanoid-derived metabolites [49]. In pineapple research, *4CL* is also an important gene that affects the synthesis of phenolic compounds. [50]. Similar to the results in citrus fruit [51], the *4CL1* and *4CL2* genes were positively correlated with the anthocyanin content of the two plums, indicating that they are also key regulatory genes in anthocyanin accumulation. *CHS* is the first key enzyme regulating the synthesis of 4-coumaroyl CoA in the direction of anthocyanin synthesis and plays a vital role in the synthesis of anthocyanins and other flavonoids [52]. In this study, the *CHS* gene was positively correlated with the flavanols and anthocyanins in the peel and was positively correlated with the total phenols, total flavonoids, and phenols in the pulp, which was consistent with the results of Ma et al. [53]. *HCT* is an acyltransferase in the chlorogenic acid biosynthetic pathway. Zhao et al. [54] demonstrated that the *HCT* gene is a key regulatory factor in the chlorogenic acid biosynthetic pathway. In this study, the expression of *HCT1* was consistent with the chlorogenic acid and neochlorogenic acid contents, suggesting that *HCT1* may be an important regulator of the accumulation of chlorogenic acid in plums. The expression of nine phenolic synthesis-related genes in CHL was higher than that in QCL. The *CHS* and *HCT1* genes were closely related to the synthesis of phenolic compounds during the fruit ripening of QCL, and the *HCT1* gene was closely related to the synthesis of phenolic compounds during the fruit ripening of CHL.

## 5. Conclusions

It was stated that the process of plum ripening had an impact on the TPC, TFC, TFAC, TMAC, phenolic components, antioxidant activities, and phenolic-metabolism-related gene expression levels of two plum cultivars. The results showed that the TSS and SS contents were highest at the mature stage of fruit development, and the contents of total phenols, total flavonoids, and total flavanols were lower than those at the young fruit stage. The total anthocyanin content was highest in CHL, and the antioxidant capacity was consistent with the changes in phenolic substances. Neochlorogenic acid, chlorogenic acid, ferulic acid, benzoic acid, rutin, and procyanidin B1 were the main phenolic components in the two plum cultivars. The results of the gene expression analysis showed that *CHS* and *HCT1* may be regulatory genes related to the accumulation of phenolic substances in QCL and CHL. *PAL3* may be a regulatory gene related to the accumulation of phenolic compounds in the pulp. *HCT1* may be an important regulator involved in the accumulation of chlorogenic acid in plums. This study provides a reference for understanding the changes in the nutritional quality and functional substances, such as phenols, in plums during fruit development and maturation. The phenolic substances in plums, particularly those in the peel, should be further utilized.

## Figures and Tables

**Figure 1 foods-11-03198-f001:**
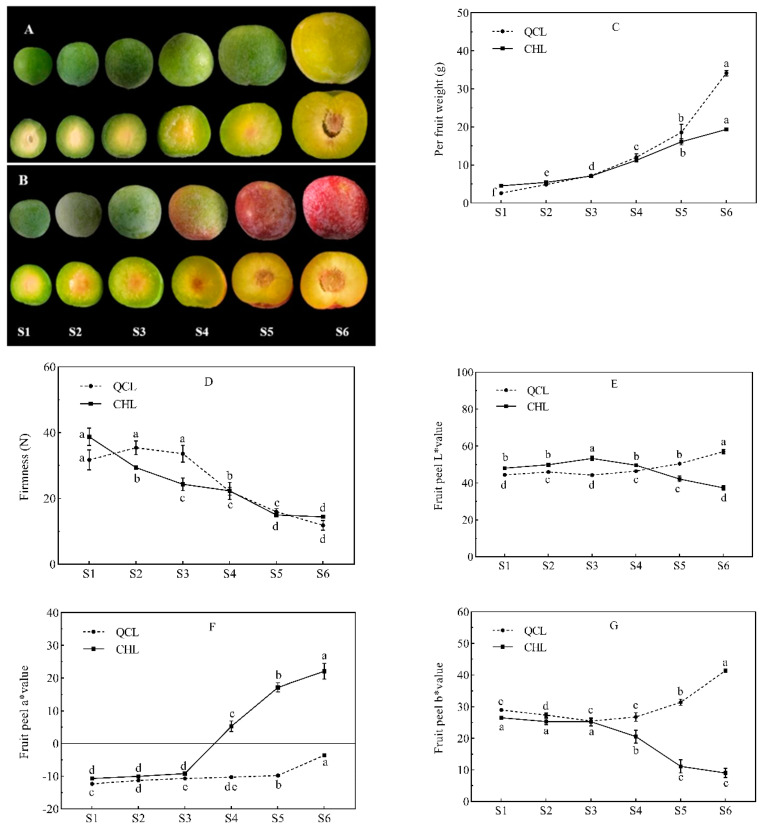
Changes in appearance quality during plum development. Photographs of the six development stages (S1–S6) of (**A**) ‘Qiangcuili’(QCL) and (**B**) ‘Cuihongli’(CHL); (**C**) fruit weight; (**D**) firmness; (**E**) fruit L* value; (**F**) fruit a* value; (**G**) fruit b* value. Developmental stages S1–S6 correspond to days 42, 56, 70, 84, 98, and 112 after anthesis. Different letters (a–f) indicate significant differences at *p* < 0.05 by Duncan’s test. Each point on the graph shows the mean and standard error, n = 3.

**Figure 2 foods-11-03198-f002:**
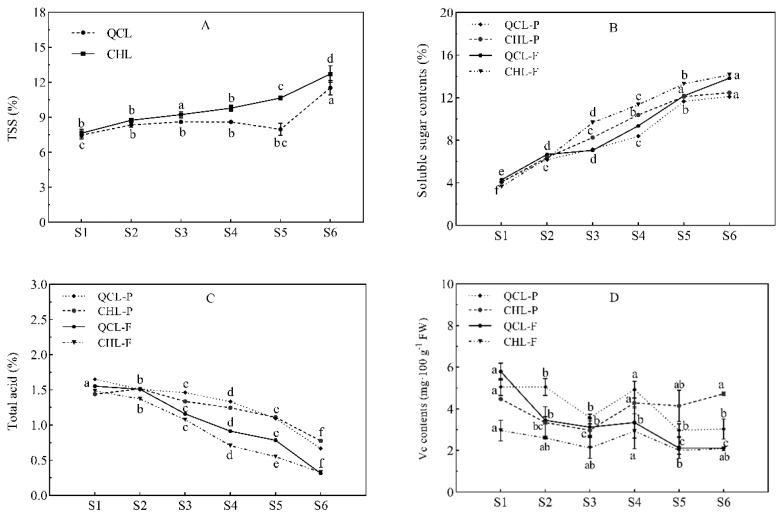
Changes in TSS (**A**), soluble sugar contents (**B**), TA contents (**C**), and Vc contents (**D**) during the development of plums. QCL-P and CHL-P represent the peel of ‘Qiangcuili’ and ‘Cuihongli’, respectively. QCL-F and CHL-F represent the flesh of ‘Qiangcuili’ and ‘Cuihongli’, respectively, the same as below. Different letters (a–f) indicate significant differences at *p* < 0.05 by Duncan’s test. Each point on the graph shows the mean and standard error, n = 3.

**Figure 3 foods-11-03198-f003:**
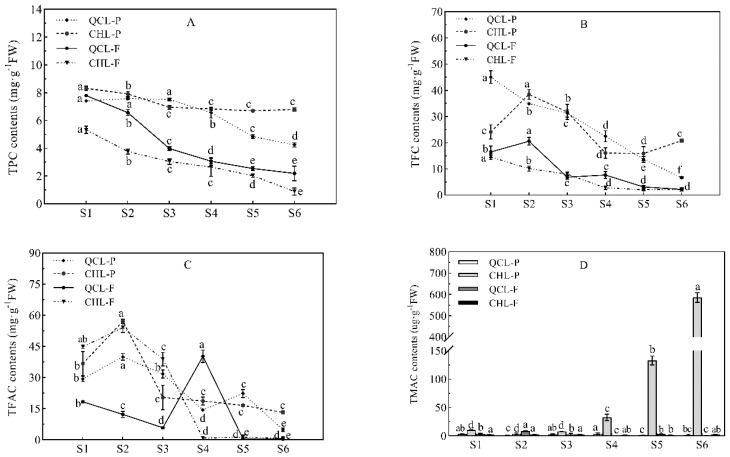
Total phenolic (TPC), total flavonoid (TFC), total flavanol (TFAC), and total anthocyanin contents (TMAC) in the peel and pulp of the plum cultivars ‘Qiangcuili’ and ‘Cuihongli’ during development. (**A**) TPC; (**B**) TFC; (**C**) TFAC; and (**D**) TMAC. Different letters (a–f) indicate significant differences at *p* < 0.05 by Duncan’s test. Each point on the graph shows the mean and standard error, n = 3.

**Figure 4 foods-11-03198-f004:**
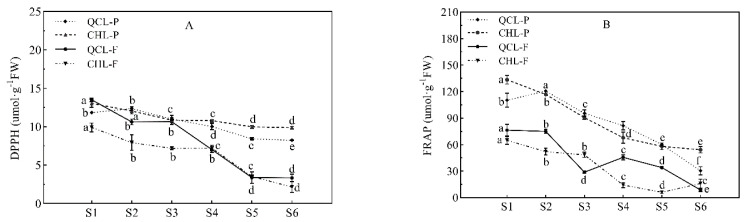
Change in antioxidant activity in the skins of plum cultivars ‘Qiangcuili’ and ‘Cuihongli’ during development. (**A**) DPPH radical scavenging capacity; (**B**) FRAP ferric reducing antioxidant capacity. Different letters (a–f) indicate significant differences at *p* < 0.05 by Duncan’s test. Each point on the graph shows the mean and standard error, n = 3.

**Figure 5 foods-11-03198-f005:**
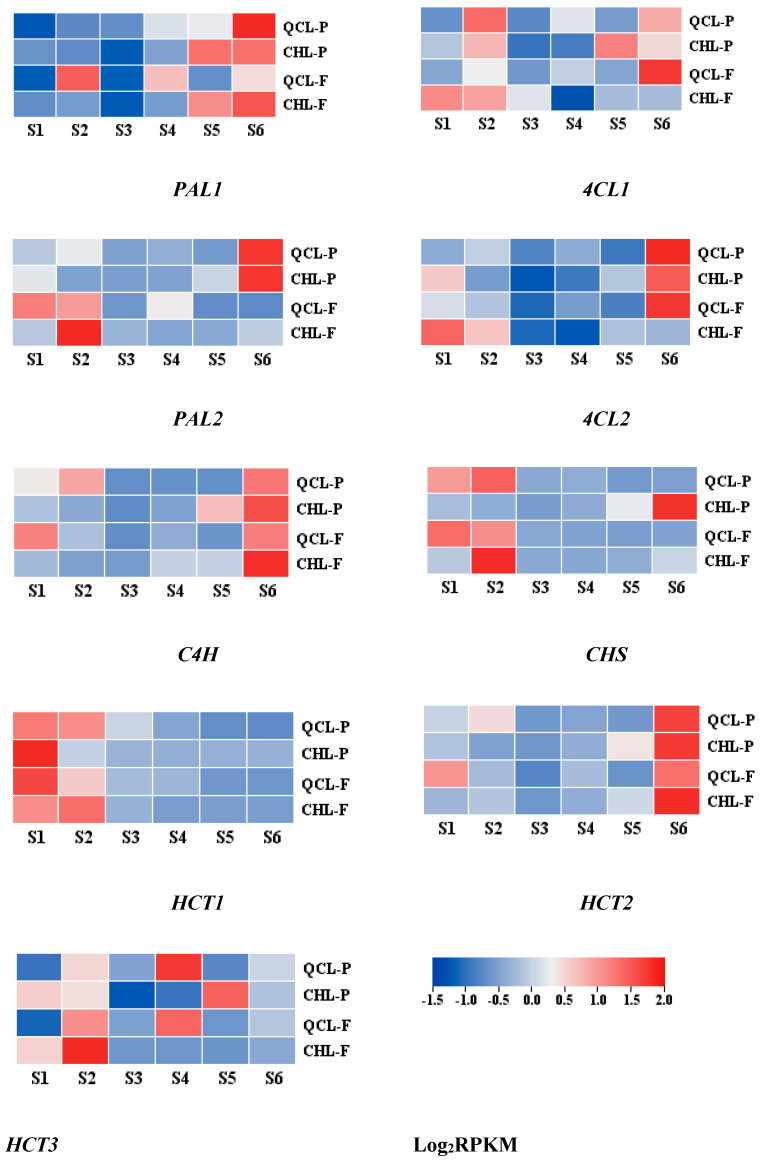
Gene expression of *PAL1*, *PAL3*, *C4H*, *4CL1*, *4CL2*, *CHS*, *HCT1*, *HCT2*, and *HCT3* during the development of QCL-P, CHL-P, QCL-F, and CHL-F.

**Table 1 foods-11-03198-t001:** Changes in gallic acid, neochlorogenic acid, chlorogenic acid, vanillic acid, ferulic acid, benzoic acid, rutin, quercetin, and procyanidin B1 during plum development.

Developmental Site	Species	Phenolic Acids (mg·100 g^−1^ FW)						Flavonol Component (mg·100 g^−1^ FW)		Flavanol Component (mg·100 g^−1^ FW)
		Gallic Acid	Neochlorogenic Acid	Chlorogenic Acid	Vanillic Acid	Ferulic Acid	Benzoic Acid	Rutin	Quercetin	Procyanidin B1
QCL-P	S1	2.10 ± 0.06 a	37.4 ± 0.79 a	51.99 ± 4.59 a	3.84 ± 0.12 b	2.19 ± 0.06 a	5.70 ± 0.18 b	3.43 ± 0.14 b	ND	174.68 ± 10.23 a
	S2	1.66 ± 0.02 b	24.87 ± 0.05 b	35.10 ± 1.41 b	35.15 ± 1.83 a	1.51 ± 0.17 b	20.22 ± 0.11 a	17.64 ± 0.55 a	15.83 ± 0.85 a	66.55 ± 4.33 d
	S3	1.07 ± 0.03 c	18.88 ± 0.55 c	24.68 ± 0.42 c	2.74 ± 0.11 bc	1.25 ± 0.02 c	4.39 ± 0.35 c	1.93 ± 0.07 cd	2.51 ± 0.08 b	145.67 ± 5.08 b
	S4	0.98 ± 0.02 c	14.19 ± 0.50 d	18.41 ± 1.62 c	0.90 ± 0.02 c	1.1 ± 0.07 cd	5.40 ± 0.19 b	1.18 ± 0.06 d	0.55 ± 0.01 c	151.13 ± 6.92 b
	S5	0.64 ± 0.03 d	8.70 ± 0.42 e	10.60 ± 0.33 d	1.5 ± 0.02 bc	0.88 ± 0.01 de	4.12 ± 0.20 c	1.61 ± 0.04 cd	1.31 ± 0.01 c	97.64 ± 3.48 c
	S6	0.50 ± 0.00 e	5.18 ± 0.32 f	7.82 ± 0.37 d	0.82 ± 0.03 c	0.70 ± 0.03 e	3.28 ± 0.03 d	2.06 ± 0.20 c	0.70 ± 0.04 c	65.30 ± 4.59 d
CHL-P	S1	4.44 ± 0.08 a	3.44 ± 0.20 b	5.63 ± 0.11 c	ND	1.05 ± 0.04 c	11.28 ± 0.21 b	3.13 ± 0.27 c	ND	5.63 ± 0.01 c
	S2	4.19 ± 0.11 b	3.47 ± 0.15 b	5.29 ± 0.16 c	0.90 ± 0.02 b	1.40 ± 0.13 ab	10.89 ± 0.46 b	2.55 ± 0.15 c	ND	5.29 ± 0.01 d
	S3	2.89 ± 0.10 c	2.77 ± 0.10 c	10.60 ± 0.33 a	ND	1.34 ± 0.04 b	4.98 ± 0.13 d	11.33 ± 1.53 b	ND	10.60 ± 0.02 a
	S4	2.06 ± 0.02 d	1.47 ± 0.14 d	3.16 ± 0.31 d	ND	0.47 ± 0.01 d	6.70 ± 0.64 c	2.05 ± 0.04 c	ND	3.15 ± 0.02 f
	S5	ND	3.36 ± 0.19 b	3.65 ± 0.10 d	ND	1.57 ± 0.02 a	16.97 ± 0.85 a	10.66 ± 0.62 b	10.05±0.47 a	3.67 ± 0.05 e
	S6	ND	7.95 ± 0.05 a	8.07 ± 0.28 b	18.67 ± 1.36 a	1.34 ± 0.04 b	18.37 ± 0.53 a	17.07 ± 0.78 a	4.86 ± 0.29 b	8.02 ± 0.03 b
QCL-F	S1	ND	24.79 ± 2.66 a	14.96 ± 0.79 a	5.51 ± 0.31 a	1.94 ± 0.06 a	8.68 ± 0.23 c	3.34 ± 0.07 b	ND	52.02 ± 7.18 a
	S2	ND	20.76 ± 0.39 b	5.12 ± 0.29 b	3.69 ± 0.10 b	1.46 ± 0.07 b	19.59 ± 0.57 a	7.15 ± 0.29 a	ND	33.02 ± 0.79 b
	S3	ND	9.89 ± 1.25 c	4.10 ± 0.15 b	1.03 ± 0.03 c	1.15 ± 0.04 c	12.43 ± 0.41 b	1.68 ± 0.03 c	ND	26.33 ± 1.93 bc
	S4	ND	7.08 ± 0.76 c	2.92 ± 0.31 c	0.94 ± 0.01 c	0.83 ± 0.04 d	7.46 ± 0.08 d	ND	ND	20.35 ± 2.70 c
	S5	ND	2.99 ± 0.55 d	1.64 ± 0.19 d	ND	ND	3.71 ± 0.33 e	ND	ND	10.36 ± 1.56 d
	S6	ND	2.68 ± 0.13 d	0.91 ± 0.04 de	ND	ND	ND	ND	ND	6.70 ± 0.35 de
CHL-F	S1	ND	1.74 ± 0.31 a	1.41 ± 0.06 a	0.83 ± 0.13 a	0.73 ± 0.02 b	5.10 ± 0.17 a	4.46 ± 0.17 a	ND	4.40 ± 0.22 d
	S2	ND	1.48 ± 0.13 b	1.05 ± 0.05 b	1.16 ± 0.01 b	0.44 ± 0.01 c	3.50 ± 0.26 b	2.02 ± 0.12 b	ND	4.32 ± 0.04 d
	S3	ND	ND	ND	ND	1.30 ± 0.03 a	ND	1.49 ± 0.05 c	ND	35.38 ± 0.01 a
	S4	ND	ND	ND	ND	ND	ND	ND	ND	6.70 ± 0.12 c
	S5	ND	ND	ND	ND	ND	ND	ND	ND	11.64 ± 0.01 b
	S6	ND	2.10 ± 0.20 b	ND	ND	ND	ND	ND	ND	ND

Values of three replicates are expressed as means ± SD. Different lowercase letters (a–f) in columns denote significant differences between sampling dates for each cultivar by Duncan’s multiple range test (*p* < 0.05) (ND, not detected). QCL-P and CHL-P represent the fruit peel of ‘Qiangcuili’ and ‘Cuihongli’, respectively. QCL-F and CHL-F represent the flesh of ‘Qiangcuili’ and ‘Cuihongli’, respectively.

**Table 2 foods-11-03198-t002:** Pearson’s correlation coefficients of phenolics (TPC, TFC, TFAC, and TMAC) and antioxidant capacity (via DPPH and FRAP) in the peel and pulp of ‘Qiangcuili’ and ‘Cuihongli’ plums.

		TPC	TFC	TFAC	TMAC
QCL-P	DPPH	0.950 **	0.936 **	0.824 **	0.094
	FRAP	0.908 **	0.495 *	0.770 **	−0.582 *
CHL-P	DPPH	0.891 **	0.796 **	0.307	0.489 *
	FRAP	0.913 **	0.839 **	0.746 **	0.395
QCL-F	DPPH	0.952 **	0.935 **	0.888 **	0.107
	FRAP	0.938 **	0.632 **	0.798 **	−0.610 **
CHL-F	DPPH	0.894 **	0.929 **	0.428	0.660 **
	FRAP	0.844 **	0.972 **	0.944 **	0.588 *

Note: “*” denotes significant differences between various indicators by Duncan’s multiple range test (*p* < 0.05). ”**” denotes significant differences between various indicators by Duncan’s multiple range test (*p* < 0.01). QCL-P and CHL-P represent the fruit peel of ‘Qiangcuili’ and ‘Cuihongli’, respectively. QCL-F and CHL-F represent the flesh of ‘Qiangcuili’ and ‘Cuihongli’, respectively.

**Table 3 foods-11-03198-t003:** Correlation analysis of phenolic content and related gene expression during fruit development of ‘Qiangcuili’ and ‘Cuihongli’.

		*PAL1*	*PAL3*	*C4H*	*4CL1*	*4CL2*	*CHS*	*HCT1*	*HCT2*	*HCT3*
QCL−P	TPC	−0.819 **	−0.498 *	−0.161	−0.124	−0.525 *	0.617 **	0.697 **	−0.435	0.054
	TFC	−0.868 **	−0.438	−0.049	−0.233	−0.488 *	0.720 **	0.793 **	−0.358	−0.193
	TFAC	−0.793 **	−0.517 *	−0.091	−0.07	−0.591 **	0.680 **	0.703 **	−0.409	−0.25
	TMAC	−0.068	−0.224	−0.278	−0.353	−0.136	−0.132	0.061	−0.243	−0.315
CHL−P	TPC	−0.494	−0.171	−0.374	0.036	0.080	−0.355	0.827 **	−0.381	0.220
	TFC	−0.603	−0.316	−0.454	−0.019	−0.266	−0.352	0.096	−0.433	−0.173
	TFAC	−0.488	−0.365	−0.440	0.218	−0.131	−0.442	0.415	−0.484	0.253
	TMAC	0.745 **	0.910 **	0.891 **	0.336	0.744 **	0.988 **	−0.299	0.959 **	0.038
QCL−F	TPC	−0.069	0.857 **	0.285	−0.336	−0.069	0.958 **	0.936 **	0.182	−0.136
	TFC	0.237	0.888 **	0.131	−0.228	−0.11	0.916 **	0.813 **	0.074	0.199
	TFAC	0.245	0.557 **	−0.095	−0.233	−0.227	0.15	0.256	−0.041	0.573 *
	TMAC	0.288	0.504 *	−0.182	−0.175	−0.205	0.691 **	0.451	−0.25	0.169
CHL−F	TPC	−0.646	0.269	−0.655	0.453	0.586 *	0.249	0.774 **	−0.573	0.565 *
	TFC	−0.575	0.398	−0.508	0.633**	0.683 **	0.369	0.858 **	−0.391	0.643 **
	TFAC	−0.615	0.621 **	−0.590	0.634**	0.533 *	0.590 *	0.849 **	−0.427	0.750 **
	TMAC	−0.163	0.300	−0.006	0.507 *	0.331	0.339	0.427	0.151	0.451

Note: “*” denotes significant differences between various indicators by Duncan’s multiple range test (*p* < 0.05). ”**” denotes significant differences between various indicators by Duncan’s multiple range test (*p* < 0.01). QCL−P and CHL−P represent the fruit peel of ‘Qiangcuili’ and ‘Cuihongli’, respectively. QCL−F and CHL−F represent the flesh of ‘Qiangcuili’ and ‘Cuihongli’, respectively.

**Table 4 foods-11-03198-t004:** Correlation analysis of chlorogenic acid, neochlorogenic acid, and *HCT* expression in the peel and pulp of ‘Qiangcuili’ and ‘Cuihongli’.

		*HCT1*	*HCT2*	*HCT3*
QCL−P	NCA	0.853 **	−0.598 **	0.066
	CA	0.857 **	−0.549 **	0.095
CHL−P	NCA	−0.071	0.892 **	0.167
	CA	−0.084	0.099	−0.444
QCL−F	NCA	0.763 **	−0.534 **	0.121
	CA	0.964 **	−0.371	0.165
CHL−F	NCA	0.753 **	−0.339	0.290
	CA	0.841 **	−0.325	0.331

Note:”**” denotes significant differences between various indicators by Duncan’s multiple range test (*p* < 0.01). QCL−P and CHL−P represent the fruit peel of ‘Qiangcuili’ and ‘Cuihongli’, respectively. QCL−F and CHL−F represent the flesh of ‘Qiangcuili’ and ‘Cuihongli’, respectively.

## Data Availability

The data and materials supporting the conclusions of this study are included within the article.

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
