# Peer review of "Changes in Phenolic Compounds and Antioxidant Activity during Development of ‘Qiangcuili’ and ‘Cuihongli’ Fruit"

_foods, 2022, doi:10.3390/foods11203198_

Round 1

Reviewer 1 Report

I send a review of manuscript of the authors: Huifen Zhang, Jing Pu, Yan Tang, Miao Wang, Kun Tian, Yongqing Wang, Xian Luo and Qunxian Deng Changes in Phenolic Compounds and Antioxidant Activity during Development of ‘Qiangcuili’ and ‘Cuihongli’ Fruit”.

I admit that the manuscript is interesting and I read it with interest. I think that the authors should make a minor revision.

1. Introduction:

Generally:

    Line numbering is generally missing throughout in the manuscript.

Page 2 – The aim of the work - In this study, the changes in ap-pearance = (appearance) and internal quality ?… -  What do the authors mean by this? (it is not clearly specified). Could the Authors explain it and correct the aim of the work?

I request the Authors to provide more convincing arguments of the advisability of undertaking the presented research.

2. Materials and methods:

I would like to ask the authors to state how they determined hardness (3. Results - … The hardness of the two plum fruits (Figure 1D) … /or/ firmness (Figure 1D) in the investigated samples?

2.2. Determination of appearance quality

 - At least 30 fruits per cultivar were… at each stage of growth?

 - Single fruit weight was measured with an electronic balance (please specify name, type, manufacturer)

2.3. Determination of total soluble solids (TSS), soluble sugars (SS), titratable acidity (TA), and Vitamin C (Vc) content

- with the unit of %. (please put the unit in brackets)

- …by anthrone col-orimetry [18]

3. Results

3.2. Changes in TSS, SS, TA, and Vc contents during fruit development of the two plum cultivars

e.g. …and the highest Vc content was in the CHL peel (4.72 mg/100 g·FW). Please correct it to:

 (4.72 mg/100 g FW or 4.72 mg·100 g-1 FW = Please correct it throughout the 3. Results

 5. Conclusions

- The temporal dynamics of fruit quality…?- What does it mean? It is not clearly specified. (I suggest you write - it was stated that the process (stage) of plum ripening had an impact on the…

- This study provides a reference for understanding the changes in nutritional quality…? What does it mean? It is not clearly specified.

Tables

Please insert commas between the names of acids in the title of Table 1.

Please include a legend with explanations of abbreviations under Tables 2-4.

Figures

Please include under Figures - standard deviation (± SD); n=?

Author Response

Response to Reviewer 1 Comments

Thank you for taking time out of your busy schedule to review the manuscript. Now we have carefully and replied the manuscript for this revision instructions are as follows:

Point 1: Line numbering is generally missing throughout in the manuscript.

Response 1: Thank you for this valuable feedback. We added line numbers to each line of the full text.

Point 2: Page 2 – The aim of the work - In this study, the changes in ap-pearance = (appearance) and internal quality ?… -  What do the authors mean by this? (it is not clearly specified). Could the Authors explain it and correct the aim of the work?

Response 2: We are extremely grateful to you for pointing out this problem. We have explained it accordingly as following:

“Because the determination of appearance and internal quality helps us to understand and evaluate the quality of the two varieties of plum fruit more comprehensively, the first part of the discussion of the results has been analyzed and discussed.”

Point 3: I would like to ask the authors to state how they determined hardness (3. Results - … The hardness of the two plum fruits (Figure 1D) … /or/ firmness (Figure 1D) in the investigated samples?

Response 3: Thank you for pointing this out. We have revised it accordingly as following (Line 82, clean version of manuscript):

“Hardness was determined by WDGY-4 fruit hardness tester.”

Point 4: At least 30 fruits per cultivar were… at each stage of growth?

Response 4: Yes, In our experiments, more than 30 plum fruits were sampled at each sampling stage of the two plum varieties.

Point 5: Single fruit weight was measured with an electronic balance (please specify name, type, manufacturer)

Response 5: Thank you for pointing this out. We have revised it accordingly as following (Line 81, clean version of manuscript):

“Single fruit weight was measured with an electronic balance (ATX124, Made in the Phil-ippines).”

Point 6: with the unit of %. (please put the unit in brackets)

Response 6: Thank you for your suggestions. We have revised it accordingly as following (Line 86-87, clean version of manuscript):

“Determination of total soluble solids (%), soluble sugars (%), titratable acidity (%), and Vitamin C content.”

Point 7: 3.2. Changes in TSS, SS, TA, and Vc contents during fruit development of the two plum cultivars. e.g. …and the highest Vc content was in the CHL peel (4.72 mg/100 g·FW). Please correct it to: (4.72 mg/100 g FW or 4.72 mg·100 g-1 FW = Please correct it throughout the 3. Results.

Response 7: Thank you for your nice comments on our article. According to your suggestions, we have revised it accordingly as following (Line 250-252, 283-290, 338-362, clean version of manuscript):

“At the maturation (S6), the Vc content in the peel was higher than that in pulp between the two cultivars, and the highest Vc content was in the CHL peel (4.72 mg·100 g-1 FW). TPC at the fruit ripening stage was in the order of CHL-P (6.80 mg·g-1 FW) > QCL-P (4.24 mg·g-1 FW) > QCL-F (2.19 mg·g-1 FW) > CHL-F (0.88 mg·g-1 FW).etc. ”

Point 8: The temporal dynamics of fruit quality…?- What does it mean? It is not clearly specified. (I suggest you write - it was stated that the process (stage) of plum ripening had an impact on the…

Response 8: Thank you for your suggestions. We have revised it accordingly as following (Line 609-610, clean version of manuscript):

“It was stated that the process of plum ripening had an impact on the TPC, TFC, TFAC, TMAC, phenolic components, antioxidant activities, and phenolic metabolism-related gene expression levels were analyzed during the fruit development of two plum cultivars.”

Point 9: This study provides a reference for understanding the changes in nutritional quality…? What does it mean? It is not clearly specified.

Response 9: Thank you for pointing this out. In this study, the nutritional quality of TSS, TA, SS, Vc and other nutrients during the fruit development of two plum varieties were measured and analyzed, which can provide some reference for future research.

Point 10: Please insert commas between the names of acids in the title of Table 1.

Response 10: We are extremely grateful to you for pointing out this problem. We have revised it accordingly as following (Line 361-362, clean version of manuscript):

“Table 1. Changes in gallic acid, neochlorogenic acid, chlorogenic acid, vanillic acid, ferulic acid, benzoic acid, rutin, quercetin and procyanidin B1 during plum fruit development.”

Point 11: Please include a legend with explanations of abbreviations under Tables 2-4.

Response 11: Thank you for this valuable feedback. We have revised it accordingly as following (Line 408-410, 500-502, 529-521, clean version of manuscript):

“Note: “*”denotes significant differences between various indicators by Duncan’s multiple range test (p < 0.05).“**” denotes significant differences between various indicators by Duncan’s multiple range test (p < 0.01). The QCL-P and CHL-P represent the fruit peel of ‘Qiangcuili’ and ‘Cuihongli’, respectively. The QCL-F and CHL-F represent the flesh of ‘Qiangcuili’ and ‘Cuihongli’, respectively.”

Point 12: Please include under Figures - standard deviation (± SD); n=?

Response 12: Thank you for your suggestions. We have revised it accordingly as following (Line 236, 274-275, 325, 393-394, clean version of manuscript):

“Each point on the graph shows the mean and standard error, n = 3.”

To sum up, we have carefully studied your opinion and revised our manuscript. Please do not hesitate to contact us if there are any question. Thanks again to the reviewers and editors for your hard work! Best wishes to you!

Author: QunXian Deng

October 10, 2022

Reviewer 2 Report

Understanding the changes in Phenolic Compounds and Antioxidant Activity is important for the development of food products/value added products. The present study is interesting and fits with the scope of the journal. However, the following specific points should be addressed before acceptance

Abstract

Abstract section was well written. It has highlighted the background, experimental plan and its results.

Keywords: Avoid the words used in the title

Introduction

Provide the data on production of plum during 2021-2022 in china

Highlight the products/value added products derived from plum

Write the novelty of this study before objectives

Materials and methods

Why the authors have selected six different stages between 42 to 112 days? Is it the standard matrutity of the fruit? The maximum matrutiy at 112 days?

Why the authors have not studied the textural characteristics of the fruits during these six stages?

Results and discussion

The figures quality is poor. Please provide the high resolution image

I recommend the authors to add more discussion/scientific reason.

PCA test is useful to classify the category of the fruits. The authors can add the PCA along with correlation

Reference

Please update the old reference (published before 2016) with recent references

Author Response

Response to Reviewer 2 Comments

Thank you for taking time out of your busy schedule to review the manuscript. Now we have carefully and replied the manuscript for this revision instructions are as follows:

Point 1: Keywords: Avoid the words used in the title

Response 1: Thank you for pointing this out. We have screened the key words that best reflect this study.

Point 2: Provide the data on production of plum during 2021-2022 in china

Response 2: Thank you for your suggestions. We have revised it accordingly as following (Line 57, clean version of manuscript):

“By 2021, China 's plum planting area has exceeded 1,900,000 hectares (hm²).”

Point 3: Highlight the products/value added products derived from plum

Response 3: Thank you for pointing this out. We have revised it accordingly as following (Line 55-57, clean version of manuscript):

“In China, plums can not only be eaten directly, but also be processed into by-products, such as fruit juice, fruit wine, sauce and moon cake, etc.”

Point 4: Write the novelty of this study before objectives

Response 4: Thank you for pointing this out. In the preface, we first introduced the nutritional and health value of fruits and vegetables in the diet, then introduced the rich phenolic substances and antioxidant activity in the plum fruit, and finally introduced the research significance of the crisp plum varieties which are less studied at present.

Point 5: Why the authors have selected six different stages between 42 to 112 days? Is it the standard matrutity of the fruit? The maximum matrutiy at 112 days?

Response 5: Thank you for your suggestions. Yes, the fruits of the two plum cultivars reach commercial maturity about 112 days after anthesis, that is, the fruit ripening harvest time.

Point 6: Why the authors have not studied the textural characteristics of the fruits during these six stages?

Response 6: Thank you for your suggestions. We reflected the fruit texture by hardness index.

Point 7: The figures quality is poor. Please provide the high resolution image

Response 7: Thank you for pointing this out. We have replaced all the pictures.

Point 8: I recommend the authors to add more discussion/scientific reason.

Response 8: Thank you for your nice comments on our article. According to your suggestions,  we have added the following corresponding (Line 601-603, clean version of manuscript): In the discussion section we have also updated recent research to scientific reason.

“In pineapple research, 4CL is also an important gene that affects the synthesis of phenolic compounds. [49].”

“49. Léchaudel, M.; Darnaudery, M.; Joët, T.; Fournier, P.; Joas, J. Genotypic and environmental effects on the level of ascorbic acid, phenolic compounds and related gene expression during pineapple fruit development and ripening. Plant Physiol Biochem. 2018, 130, 127-138. doi: 10.1016/j.plaphy.2018.06.041.”

Point 9: PCA test is useful to classify the category of the fruits. The authors can add the PCA along with correlation

Response 9: Thank you for this valuable feedback. The results of correlation analysis between phenols and antioxidant activity were shown in Table 2.

Point 10: Please update the old reference (published before 2016) with recent references

Response 10: We are extremely grateful to you for pointing out this problem. I ' ve revised some of the literature, but I ' m sorry, some documents can 't be replaced before 2016, I will pay attention to this later.

  1. Chen, C.; Chen, H.; Yang, W.; Li, J.; Tang, W.; Gong, R. Transcriptomic and Metabolomic Analysis of Quality Changes during Sweet Cherry Fruit Development and Mining of Related Genes. Int J Mol Sci. 2022, 23, 7402. doi: 10.3390/ijms23137402.
  2. Choi, M.; Sathasivam, R.; Nguyen, B.V.; Park, N.I.; Woo, S.H.; Park, S.U. Expression Analysis of Phenylpropanoid Pathway Genes and Metabolomic Analysis of Phenylpropanoid Compounds in Adventitious, Hairy, and Seedling Roots of Tartary Buckwheat. Plants (Basel). 2021, 11, 90. doi: 10.3390/plants11010090.
  3. Munekata, P.E.S.; Yilmaz, B.; Pateiro, M.; Kumar, M.; Domínguez, R.; Shariati, M.A.; Hano, C.; Lorenzo, J.M. Valorization of by-products from Prunus genus fruit processing: Opportunities and applications. Crit Rev Food Sci Nutr. 2022, 14, 1-16. doi: 10.1080/10408398.2022.2050350.
  4. Li, D.; Zhang X.; Li, L.; Aghdam, M.S.; Wei, X.; Liu, J.; Xu, Y.; Luo, Z. Elevated CO2 delayed the chlorophyll degradation and anthocyanin accumulation in postharvest strawberry fruit. Food Chem. 2019, 285, 163-170. doi: 10.1016/j.foodchem.
  5. Sun, Y.; Li , M.; Mitra, S.; Hafiz, M.R.; Debnath, B.; Lu, X.; Jian, H.; Qiu, D. Comparative Phytochemical Profiles and Antioxidant Enzyme Activity Analyses of the Southern Highbush Blueberry (Vaccinium corymbosum) at Different Developmental Stages. Molecules. 2018 , 23, 2209. doi: 10.3390/molecules23092209.
  6. Chen, Q.; Wang, D.; Tan, C.; Hu, Y.; Sundararajan, B.; Zhou, Z. Profiling of Flavonoid and Antioxidant Activity of Fruit Tissues from 27 Chinese Local Citrus Cultivars. Plants (Basel). 2020, 9, 196. doi: 10.3390/plants9020196.
  7. Rahim, M.A.; Busatto, N.; Trainotti, L. Regulation of anthocyanin biosynthesis in peach fruits. Planta. 2014, 240, 913-29. doi: 10.1007/s00425-014-2078-2.
  8. Zhao, L.; Wang, D.; Liu, J.; Yu, X.; Wang, R.; Wei, Y.; Wen, C.; Ouyang, Z. Transcriptomic analysis of key genes involved in chlorogenic acid biosynthetic pathway and characterization of MaHCT from Morus alba L. Protein Expr Purif. 2019, 156, 25-35. doi: 10.1016/j.pep.2018.

To sum up, we have carefully studied your opinion and revised our manuscript. Please do not hesitate to contact us if there are any question. Thanks again to the reviewers and editors for your hard work! Best wishes to you!

Author: QunXian Deng

October 10, 2022

Reviewer 3 Report

The point is hot and could be considered for publication after fixing some issues:

No line number so the revision is hard.

- Captilaize each keword and end the sentnce with point.

- 2.3. Determination of total soluble solids (TSS), soluble sugars (SS), titratable acidity (TA), and Vitamin C (Vc) content.. how did you calculated the SS and TA by which main compenent, plz mention it?

- 2.7. Determination of total anthocyanin content.. how did you calculated the total anthocyanins from the OD you measured? and which anthocyanins you mean?

- The figure quality could be enhanced more.. all figures.

Author Response

Response to Reviewer 3 Comments

Thank you for taking time out of your busy schedule to review the manuscript. Now we have carefully and replied the manuscript for this revision instructions are as follows:

Point 1: No line number so the revision is hard.

Response 1: Thank you for your suggestions. We added line numbers to each line of the full text.

Point 2: Captilaize each keword and end the sentnce with point.

Response 2: Thank you for pointing this out. We 've made changes with reference to “Foods” keyword format requirements.

Point 3: 2.3. Determination of total soluble solids (TSS), soluble sugars (SS), titratable acidity (TA), and Vitamin C (Vc) content. how did you calculated the SS and TA by which main compenent, plz mention it?

Response 3: Thank you for your nice comments on our article. According to your suggestions, we have added the following corresponding (Line 90-97, clean version of manuscript):

“The SS content in the peel and pulp was determined by anthrone colorimetry [1819], the SS was obtained from the equation: SS (%) = C * Vt * N * 100 / Vs * W × 106 (C is sugar content obtained by standard curve, Vt is total volume of extract, N is dilution rate, Vs is sample volume at determination, W is sample weight, 106 is weight unit conversion).”

“TA was measured by sodium hydroxide titration, the TA was obtained from the equation: TA (%) = K*C*(VNaOH-V3)*V1*100/W*V2 (C is Sodium hydroxide standard titration solution concentration, VNaOH is Volume of NaOH used for titration, W is sampling amount of sample, V3 is Blank consumption of NaOH volume, V1 is constant volume, V2 is Titration volume.”

Point 4: 2.7. Determination of total anthocyanin content.. how did you calculated the total anthocyanins from the OD you measured? and which anthocyanins you mean?

Response 4: We are extremely grateful to you for pointing out this problem. We have added the following corresponding (Line 126-130, clean version of manuscript):

“The total anthocyanin concentration was obtained from the equation:

TMAC(nmol/g)=(ODλ/ελ)*(V/W)*106

Where ODλ=(OD530-OD620)-0.1(OD650-OD620), ελ is anthocyanin molar extinction coefficient 4.62*104, V is total volume of extract, M is sampling weight.”

Point 5: The figure quality could be enhanced more. all figures.

Response 5: Thank you for pointing this out. We have replaced all the pictures.

To sum up, we have carefully studied your opinion and revised our manuscript. Please do not hesitate to contact us if there are any question. Thanks again to the reviewers and editors for your hard work! Best wishes to you!

Author: QunXian Deng

October 10, 2022
